# Protocol of Breast Cancer Prevention Model with Addition of Breast Ultrasound to Routine Gynecological Visits as a Chance for an Early Diagnosis and Treatment in 25 to 49-Year-Old Polish Females

**DOI:** 10.3390/diagnostics13020227

**Published:** 2023-01-07

**Authors:** Marcin Śniadecki, Paulina Jaworek, Zuzanna Chmielewska, Patryk Poniewierza, Maria Stasiak, Martyna Danielkiewicz, Damian Stencelewski, Michał Brzeziński, Zuzanna Anna Boyke, Ewa Wycinka, Medha Sunil, Marie Nguyen, Dagmara Klasa-Mazurkiewicz, Krzysztof Koziełek, Piotr Rak, Yvonne Wolny, Marcin Liro, Paweł Władysław Guzik, Katarzyna Dobruch-Sobczak, Dariusz Wydra

**Affiliations:** 1Department of Gynecology and Obstetrics, Medical University of Gdańsk, 80-210 Gdańsk, Poland; 2The American Health Information Management Association (AHIMA), 60601-5809 Chicago, IL, USA; 3Air Chair, Medical University of Gdańsk, 80-210 Gdańsk, Poland; 4Medicover, Al. Jerozolimskie 96, 00-807 Warszawa, Poland; 5Faculty of Medicine, Lazarski University, Swieradowska 43, 02-662 Warsaw, Poland; 6Department of Gynecological Oncology, PCK Marine Hospital in Gdynia, 81-519 Gdynia, Poland; 7Department of Oncological Propedeutics, Medical University of Gdańsk, 80-210 Gdańsk, Poland; 8Department of Art History, Faculty of History, University of Gdansk, 80-309 Gdańsk, Poland; 9Department of Statistics, Faculty of Management, University of Gdansk, 81-824 Sopot, Poland; 10Mammography Laboratory, HCP Medical Center, Hospital St. John Paul II, 61-001 Poznan, Poland; 11Wyspa Medycyny Przyjaznej [The Island of Friendly Medicine], 80-747 Gdańsk, Poland; 12Amita Health St. Joseph Hospital Chicago, Chicago, IL 60657, USA; 13Clinical Department of Gynecology and Obstetrics, City Hospital, 35-241 Rzeszów, Poland; 14Ultrasound Department, Institute of Fundamental Technological Research, Polish Academy of Sciences, 02-106 Warsaw, Poland; 15Radiology Department II, Maria Sklodowska-Curie National Research Institute of Oncology, 00-001 Warsaw, Poland

**Keywords:** screening, cervical cancer, breast cancer, ultrasonography, mammography, value-based healthcare, health policy, senology

## Abstract

The low attendance rate for cancer screening tests in Poland is a major healthcare concern that requires specific analysis and the development of implementation recommendations for prevention, and both actions are likely to benefit culturally similar countries. Four female cancers account for approximately 20% of all cancer cases—breast cancer, cervical cancer, endometrial cancer, and ovarian cancer—suggesting that gynecologists have a significant preventative role. Of the four, breast cancer and cervical cancer are among the 10 most common malignant neoplasms globally, regardless of gender, occur only in women and are known to have effective screening measures. Our research aims to create a screening model that combines cervical cancer and breast cancer to maximize health outcomes for women at risk of both cancers. In the study protocol, we have created a model that maximizes benefits for patients with minimal additional costs to the health care system. To achieve the set goal, instead of regular clinical breast exams as recommended by the gynecological societies, we proposed an ultrasound examination, during which palpation may also be performed (in the absence of elastography). We present a scheme for such a protocol that takes into consideration all types of prevention in both cancers, and that emphasizes breast ultrasound as the most frequently missing element. Our study includes a discussion of the strengths and weaknesses of our strategy, and the crucial need for infrastructure and education for the successful implementation of the program. We conclude that our model merits consideration and discussion among health-care decision makers, as the screening changes we propose have significant potential benefits for the female population.

## 1. Introduction

Breast cancer (BC) is the most common cancer worldwide with 2,261,419 registered cases for both sexes (11.7%) [1]. Although the incidence of BC in the young age group (up to 35 years) is about 5%, it is expected to grow [2,3]. The estimated distribution of new cases of BC in the countries of the European Union (EU) in 2020 in age group 20–44 years was 40,613 cases [4]. Cervical cancer (CC) is the most common gynecological cancer globally and the second most common cancer occurring in women aged 15–44 years [1,5]. The estimated distribution of new cases of cervical cancer in the EU in 2020 by age group 20–44 years was 8635 cases [6].

Although the incidence rate of BC in the older age group (women up to 44) is stable or decreasing [7], for younger women (aged 44 and lower) the incidence rate is increasing. Furthermore, younger women are diagnosed with breast cancer at more advanced stages and are more likely to have a biologically aggressive phenotype, which leads to lower survival rates [8]. Not only does it generate costs associated with more aggressive treatment and hospitalization of these women [9], but it also leads to the greatest productivity loss in the female working-age population [10].

In both cancers, the time of the preclinical development phase is long enough that it is possible to identify a precancerous lesion (or low stage BC) at several time points during the screening period [11,12].

For CC, the main incident factor is known and there are tests based on ASSURED principles (by World Health Organization) to ensure early detection of high-risk human papillomavirus (hrHPV) infection [13,14]. It is reasonable to suppose that approximately 20 percent of women aged 14 to 19 in countries with a high human development index (HDI) are infected with one of the types of HPV [15]. For comparison, cases of genetically determined BC constitute only about 5% of cases; however, the call for predisposition tests is similar to that of preventive vaccinations in the case of HPV [16]. The remaining risk factors for BC are largely modifiable [17]. While in many countries for average-risk women, risk factor control as primary prevention and cyclic mammography (MMG) for women >45 years of age as secondary BC prevention, and risk factors control (together with hrHPV vaccination) and cyclic liquid-based cytology (LBC) for CC prevention are available and recommended [18], insufficient emphasis is put on reaching this population, which constitutes more than half of the female population in these countries. The latter can be seen in the Polish population where the proportion of people participating in screening is declining (see Appendix A) [19].

According to the Transparency Council of the Agency for Health Technology Assessment and Tariff System (AOTMiT), the screening program should promote combining CC prevention with BC prevention. At the same time, the Council points out that the issue of participation in preventive programs is not a biological issue, but a psychological and cultural-dependent one [20].

Screening behavior is influenced by a multitude of factors like fear of the result, embarrassment, anticipation of pain during the examination, employment status, education level, insurance coverage/accessibility, and other misconceptions like focusing on the negative side effects of screening in the media [21]. There is a spill-over effect in the psychology of screening, which is that a woman already participating in one screening program will be more likely to participate in another [22].

Failure to participate in screening tests is more often the result of unawareness and we cannot speak of a decision not to participate. Several authors pointed out that the appeal of medical personnel to participate in the second screening program, preceded by education and the combination of two programs at the same time, may increase women’s participation in each of them [22,23]. There was a very important sentence supporting the thesis in a Japanese study: “84.9% of women who participated in CCS also participated in BCS and 82.6% of women who participated in BCS also participated in CCS” [24]. An interventional study was done in Tshwane, South Africa, to see whether combining breast and cervical screening could improve cervical screening uptake. This study showed that despite CC being the most common cancer in black South African women, there was a lack of knowledge of CC compared to BC, thus combining breast and CC screening programs increased the screening uptake of CC. Awareness was created first in the form of posters, flyers, and invitations for examinations. The participants were also educated about both cancers, their signs, and symptoms as well as the screening process. This later showed that once the participant knew about the importance of screening, they were more likely to be engaged in cancer prevention [25].

In the United States, there are such programs in existence such as the National Breast and Cervical Cancer Early Detection Program (NBCCEDP) of the Centers for Disease Control and Prevention (CDC), but the screening of the two cancers is not combined; it is more like two different screening programs are funded by a single entity. In Poland, there is a general oncological program called “Planuje długie życie” [I am planning a long life] that involves cancer prevention programs and in particular BC prevention not different from the actual standard screening. Until now in Poland women aged 50 to 69 years were recommended to be screened with MMG every two years and may be prompted for re-mMMG after a year if they have risk factors such as BC in their immediate family [26]. According to recommendations that are currently in the process of implementation, the most efficient cervical cancer prevention scheme recommends every female patient aged 25 and older to have hrHPV (16 and 18) test on a 5-year basis with subsequent cytology (preferable LBC) in the case of a positive test result [27]. Performing breast ultrasound (US) is an optional screening test and is up to the physician’s and patient’s decision, however, it should be noted that the US gives high levels of sensitivity and specificity (80% and 88%, respectively) in BC detection [28]. Merging breast ultrasound with CC screening can be beneficial for younger patients that are not included in the age group for MMG checks. General lack of screening program popularity among Polish women and lack of extended research that investigates structural conditions related to breast US may be the biggest problem in achieving improvement in local healthcare as well as globally.

The strengths, weaknesses, opportunities, and threats (SWOT) analysis of such a program combining both screenings are included in Table 1. In summary, the program itself has very significant strengths and opportunities with weaknesses and threats that can easily be addressed with proper strategy. It targets groups that are excluded from mammography giving them diagnostic opportunities.

Our strategic goal was to create a model that would combine cervical and breast cancer screening in one place and time for a woman at average risk of both cancers. We wanted to create a new rule of good practice by breaking the existing compromise between value and cost. This is in line with the blue ocean strategy business thinking model [29].

The goal of this pilot study was to check how many females aged 25–49 who would never get a breast ultrasound or mammogram otherwise have early breast cancers in their breasts that exist and are not diagnosed—to identify early breast cancers.

## 2. Experimental Design

Screening strategys could be improved significantly by creating a program that involves merging screening for CC and BC. To prevent late BC detection in patients who are too young for referral mammography, which in Poland (as of 23 November 2022) is less than 50 years old, screening programs should consider patients 25–49 years old. Taking under consideration the fact that there is work underway to introduce universal access to hrHPV-DNA tests every five years with lesions not requiring a second screening level (i.e., colposcopy-guided biopsy), patients should have at least seven breast US examinations (ICD 9 code 88.732) during a 25 years period since MMG is initiated on regular basis. Based on the data from Krajowy Rejestr Nowotworów [National Cancer Registry] [30] between 2008 and 2019 a total of 207,905 patients were diagnosed with BC (ICD 10 code C50) and 38716 of the women from that group were between 25 and 49 years old which amounts to 19% of all diagnoses.

As shown in Figure 1, the number of cases increased in all groups with remarkably fast growth in the 30–50 age group. We can assume that because the increments in the 0–39 group were higher than in the 0–34 group, the 35–39 group contributed to this increase. This is confirmed by the other data provided by the Global Cancer Observatory (GLOBOCAN). (Appendix A). To successfully convince this group of patients to take part in the cancer prevention program, a health promotion campaign targeting a wide group of responders had to be designed. Figure 1 also demonstrates the total number of diagnosis and deaths associated with ICD code C50. It was shown that patients in the age group 25–49 constituted 19% of all diagnostic cases and 9% of deaths, which, taking under consideration poor diagnostic potentials for those patients, are very significant numbers.

To assure our model was within maximum values for patients, we assumed a few key elements:conducting health education in secondary schools until health awareness is obtained (parent education in primary school)educating specialists in obstetrics and gynecology (OB-GYN) in the field of genetic counseling and supplementing their knowledge of BC prevention at the basic level of this specialization—including learning how to perform a breast examination using combined ultrasound and palpationmerging primary secondary prevention of both cancers at the same time and placein the prevention of breast cancer:risk individualization with qualification for genetic testing at the first gynecological visitaddition of a routine breast ultrasound examination to the previously performed breast palpation examination, further management according to the result of the Breast Reporting Imaging and Data System (BIRADS) scale [31]in the prevention of cervical cancer:vaccination of young girls against hrHPV with at least bivalent vaccinehrHPV test every five years, followed by LBC in the case of a positive hrHPV test result

## 3. Materials and Equipment

Based on the statistical data (Figure 1) the incidence of BC (but it also refers to CC) is significant in females aged 25–49 and because of that we propose the addition of breast ultrasound examination to gynecological visits as a standard procedure.

Among BC risk factors we can differentiate genetic and non-genetic. In genetic risk factors the most common are BRCA1 and BRCA2 mutations as well as other less common: checkpoint kinase 2 (CHEK2); cadherin 1 (CDH1); PTEN; serine/threonine protein kinase 11 (STK11; also known as LKB1); TP53; CHEK2; ataxia telangiectasia mutated (ATM); nibrin (NBN); and partner and localizer of BRCA2 (PALB2) [32]. Taking into consideration that genetic testing is still very costly and not every patient in the desired age group 25–49 knows their exact family medical history, determining the level of BC risk via simple online tools and breast ultrasound during routine gynecological visits would create a surrogate for vaccination and corresponds to LBC in CC prophylaxis, respectively [33]. The detailed analysis of the source of increasing numbers of CC in young and very young females deserves another publication; one can underline that the risk factors for BC are easily identifiable and should be translated to easy-to-read educational resources.

All these data constitute the conclusion that the numbers associated with breast cancer non-genetic risk factors has grown over the years and affects the age group in interest (25–49 years old). It is highly needed for the screening methods for those patients to be evaluated, changed and implemented nationally.

To reach each responder, the campaign is designed based on the female life timeline (Figure 2) where prevention starts at the age of 9 with HPV vaccination, up to age 74 when the last routine pap smear is recommended [34].

The currently proposed BC prevention model refers to a very similar age group of patients [19]. However, it must be mentioned that women usually have their first gynecological check-up visit earlier than at the age of 25, whether for health checks, contraception consultation or because of other reasons. Therefore, the proposed BC prevention program should be implemented, simultaneously with CC prevention, at the time of the first gynecological visit and could be also called opportunistic screening. The importance of BC prevention should be introduced to girls in the early stages. The promotional strategy must also be applied to the second target group, which is doctors who will be implementers of the program. To ensure a positive response from that group, weaknesses, and threats must be addressed. According to the recommendations of the American Cancer Society (ACS) and the Polish Oncology Union (POU), in women aged 20–39 years, it is recommended to undergo a clinical breast examination (CBE). To dispel the time difference between palpation and US examination, videos serving to compare these two methods are provided. The video on palpation includes a self-examination that also defines the proper time of CBE (Appendix A). Moreover, in the US examination a maneuver was included that corresponds to the palpation of the breast (fingers in front of the US probe; see Appendix A). It is worth noting that by having elastography in US equipment it is possible to replace palpation during US with a stiffness measurement. Comparing those two methods (palpation and US), the difference in time consumption is not very significant and should not harm program implementation. Taking into consideration the financial factor, the change to the procedure code has to be mentioned and explained. After implementing the program, they would add 88.732. to their procedure codes.

### Success Story

Our patient, Anna T., is a 29-year-old female with a history of hyperprolactinemia and irregular menstrual bleeding. She underwent a gynecological visit on 17 May 2022. Due to an abnormal result of the prolactin concentration, as well as a family history, the gynecologist suggested a breast ultrasound, which was her first. Examination revealed the presence of a BI-RADS-3 lesion (suspected fibroadenoma) in the right breast, 9 mm × 6 mm × 9 mm, with mixed breast texture, ACR 3.

The next gynecological and senological visit took place on 8 November 2022. The observed lesion in the right breast (8 mm × 6 mm × 10 mm) was graded BI-RADS-4a and stiffness was 83.5 kPa in shear-wave elastography. The patient was referred for a mammotome biopsy. The mass was not palpable by the patient, and before the first visit, she was not aware of it. In the video (Appendix A) there is documentation of the right breast ultrasound. 

Thanks to the use of breast ultrasound combined with palpation and the addition of elastography, it was possible to decide whether the lesion in the right breast should be subjected to further observation or tissue biopsy. According to the current recommendations, palpation is a standard procedure in the prevention of BC for the patient’s age group, but it does not allow for a quick effective decision.

## 4. Detailed Procedure

The above-mentioned patient’s ultrasound was performed using the tested apparatus HOLOGIC-Supersonic Aixplorer Mach 20, on the linear probe L18-5 (frequency range 5–18 MHz). Most of the settings are predefined in the “breast” preset or we can set and save them ourselves. The basic principle is to capture the layering of the breast as there is a need to freeze the image for a moment and show the location of structures, which must be maintained throughout the examination. A thorough examination of the patient had already been carried out. The recorded video (Appendix A) shows the key elements of the examination as well as selected possibilities of the current ultrasound. It should be emphasized that the video should not be a single source of knowledge on how to properly perform a breast ultrasound examination. Participation in training under the supervision of an expert is recommended to successfully implement the program. The study omitted some of the changes, and the focus was on changing BI-RADS-4a since this change needs to be defined, and elastography is an additional tool here when in doubt. In shear wave elastography, the cut-off point of a suspicious lesion was set to >85 KkPa. The test was performed one week before the recording and was described in accordance with the appropriate scheme. (Appendix A).

For practical steps see Appendix A.

### Pilot Study Protocol

To test the method, we will recruit 10 gynecologists with private practices who will agree to perform for a period of 6 months breast ultrasounds for every patient aged 25–49 who did not have it done for at least a year and normally would not have it performed at the visit. The only criteria will be not having had a breast ultrasound or mammogram ever before and signing the documents confirming participation on the study agreement. This way we will be able to observe how often females are not aware of their health status and show the problem behind self-examination in females aged 25–49. Examiners will mark those patients’ data with a special label and after 6 months we will apply the results.

## 5. Expected Results

The strategy explained in this model protocol is expected to bring positive outcomes on breast cancer diagnosis in patients excluded from the other breast cancer prevention programs as well as the overall growth of the popularity of female cancer prevention screening. Combining two programs and advertising them as a strategy for a healthy lifestyle will possibly convince females that their health is very important, and it does not have to be expensive and time-consuming. Currently, in Poland, there are different strategies and recommendations like “Planuje długie życie” (I am planning a long life) or “Dzień na U” (Day for Ultrasound) but none of them seem to address all of the issues that are associated with health promotion strategies starting with doctors’ training, patients’ lack interest, up to advertising strategies that ensure its effectiveness. Presentation of the program in the form of a timeline with the association between different age groups will educate females that at every stage of their lives they are responsible for either themselves, their daughters, or their mothers. This type of prevention program, if successfully introduced and supervised, could bring an opportunity for efficiently selecting groups for genetic testing and in the future could significantly reduce the costs of breast cancer treatment in females from all age groups. The results of this pilot study will be validated in a randomized study.

## Figures and Tables

**Figure 1 diagnostics-13-00227-f001:**
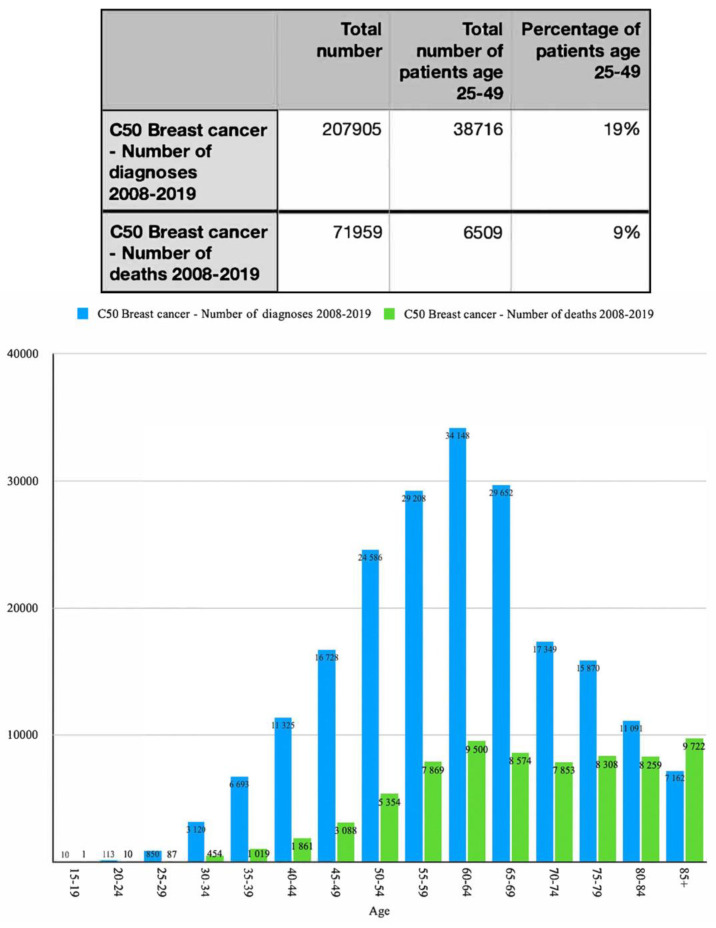
Number of breast cancer diagnoses and deaths between 2008 and 2019 in different age groups.

**Figure 2 diagnostics-13-00227-f002:**
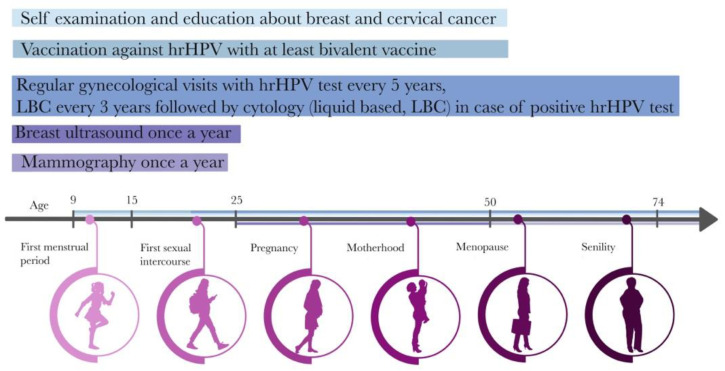
Healthy female timeline.

**Table 1 diagnostics-13-00227-t001:** The strengths, weaknesses, opportunities, and threats (SWOT) analysis of introducing the program.

STRENGTHS Targets group excluded from mammography One visit with two possible diagnostic potentials Time efficient-breast palpation with the addition of ultrasound is not much more time consuming Easier to convince a patient to do 2 checkups together than go to 2 separate visits	WEAKNESSES Costs-addition of ICD 88.732 to the visit Training of the doctors in performing breast palpation with ultrasound General lack of popularity of screening programs
OPPORTUNITIES Cancer prevention education among younger patients Women in the age group 25–49 often use birth control or come to visits after labor, before and during pregnancy so their gynecological visits are somehow routine- easy to convince them that additional tests are necessary The growing popularity of birth control in Poland over the years may be a helpful tool to convince patients to check for cervical cancer and breast cancer—a mandatory visit with checkup every year before a prescription refill For patients with disabilities that limit their ability to stand, ultrasound can potentially be the only screening method used.	THREATS Lack of physician’s training The old generation of equipment in the medical offices Women from older groups that should receive mammography may be preferring ultrasound instead because of general aversion towards mammography Lack of extended research regarding ultrasound diagnostic potential

## Data Availability

Not applicable.

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
