# Peer review of "Protocol of Breast Cancer Prevention Model with Addition of Breast Ultrasound to Routine Gynecological Visits as a Chance for an Early Diagnosis and Treatment in 25 to 49-Year-Old Polish Females"

_diagnostics, 2023, doi:10.3390/diagnostics13020227_

Round 1

Reviewer 1 Report

The authors try to create a breast cancer prevention model with the addition of breast ultrasound to routine gynecological visits as a chance for an early diagnosis and treatment in 25-49 years old.

1) The topic should be important. The authors should show the goal of this pilot study. Actually, what is a goal?  How do the authors compare the results for supporting your trial?  

2) Also, the authors should estimate statistical analysis for judging study validation or, show the next step (e.g. randomised study project)

3) Abstract: line39-47 may not be necessary because those are not the contents of the study. 

4) The sentences in Scheme and Figures are small.

5) study protocol: The authors should add the detail of inclusion criteria (e.g. informed consent, past history). The explanation is very small, but very critical for this manuscript because the type of paper is "protocol". 

Author Response

Reviewer 1 comments

Dear Reviewer, thank you for your time and valuable remarks.

“The authors try to create a breast cancer prevention model with the addition of breast ultrasound to routine gynecological visits as a chance for an early diagnosis and treatment in 25-49 years old.”

  • “The topic should be important. The authors should show the goal of this pilot study. Actually, what is a goal?  How do the authors compare the results for supporting your trial?”  

Thank you for this key remark.

“The goal of this pilot study is to check how many females aged 25-49 that would never get the breast ultrasound or mammogram otherwise have changes in their breasts that exist and are not diagnosed.”

The above passage is now included in the introduction section.

  • “Also, the authors should estimate statistical analysis for judging study validation or, show the next step (e.g. randomised study project)”

Thank you for this valuable comment. At this stage of the study design (not yet in the implementation phase), we would consider it is reasonable to only indicate that the results of the pilot study will be validated in the course of the randomized trial.

So, the dedicated phrase could be as follows:

Results of this pilot study will be validated in a randomized study. (The location of this sentence is at the end of the Expected results section).

  • “Abstract: line39-47 may not be necessary because those are not the contents of the study.”

Thank you for this comment. We removed these text lines to make the article more clear to the Reader according to suggestion.

  • “The sentences in Scheme and Figures are small.”

This is fixed according to your suggestion as well as this of another Reviewer.

  • “study protocol: The authors should add the detail of inclusion criteria (e.g. informed consent, past history). The explanation is very small, but very critical for this manuscript because the type of paper is "protocol".”

Thank you for this key remark. We think that “the only criteria will be not having breast ultrasound or mammogram ever before and signing the documents confirming participation on the study agreement.”

Reviewer 2 Report

The authors detail a need to include regular ultrasound BC screening alongside typical CC screenings for age 25-49, prior to the currently recommended mammography screenings after age 50.

Figure 1. Could the style of the chart be condensed due to the overlapping nature of the data. Font sizes increased.

Figure 2. Are both the table and graph necessary since they display the same information? The final three columns of table could be kept. Also adjust the positioning of the data above the graph bars for easier reading.

Scheme 1 Figure. Typo, should be “least” bivalent vaccine. Also please increase font size.

Line 219. Please define OB-GYN

Line 230. Please define BIRADS

Author Response

Reviewer 2 comments

Thank you for your comments. Please find answers to your comments below.

“The authors detail a need to include regular ultrasound BC screening alongside typical CC screenings for age 25-49, prior to the currently recommended mammography screenings after age 50.”

“Figure 1. Could the style of the chart be condensed due to the overlapping nature of the data. Font sizes increased.”

Thank you for this valuable remark. After rethinking, we decided to remove Figure 1. Figure 2 replaced it. We made font adjustments.

“Figure 2. Are both the table and graph necessary since they display the same information? The final three columns of table could be kept. Also adjust the positioning of the data above the graph bars for easier reading.”

Thank you for this valuable remark. We reorganized the layout of Figure 2 based on your comments.

“Scheme 1 Figure. Typo, should be “least” bivalent vaccine. Also please increase font size.”

Thank you for these suggestions. We changed Scheme 1 to Figure 2. We followed the suggested changes.

“Line 219. Please define OB-GYN”

Thank you for your remark. We defined OB-GYN in abbreviation list as well as in the manuscript text.

“Line 230. Please define BIRADS”

Thank you for your remark. We defined BIRADS in abbreviation list as well as in the manuscript text.

In addition, we reordered the abbreviation list.

Round 2

Reviewer 1 Report

The comments and corrections are reasonable.

Author Response

Thank you for accepting our responses to Reviewer's comments.